# Spatiotemporal patterns of anemia among lactating mothers in Ethiopia using data from Ethiopian Demographic and Health Surveys (2005, 2011 and 2016)

**Alemneh Mekuriaw Liyew** [1]*, **Sewnet Adem Kebede**[1], **Chilot Desta Agegnehu** [2], **Achamyeleh Birhanu Teshale**[1], **Adugnaw Zeleke Alem**[1], **Yigizie Yeshaw**[1,3], **Getayeneh Antehunegn Tesema**[1]

1 Department of Epidemiology and Biostatistics, Institute of Public Health, College of Medicine and Health Sciences and Comprehensive Specialized Hospital, University of Gondar, Gondar, Ethiopia, 2 School of Nursing, College of Medicine and Health Sciences and Comprehensive Specialized Hospital, University of Gondar, Gondar, Ethiopia, 3 Department of Physiology, School of Medicine, College of Medicine and Health Sciences Comprehensive Specialized Hospital, University of Gondar, Gondar, Ethiopia

* alemnehmekuriawliyew@gmail.com

**Data Availability Statement:** As Ethiopian demographic and health survey is part of demographic and health survey (DHS), it is publicly

## Abstract

### Introduction

Maternal anemia is a worldwide public health problem especially in developing countries including Ethiopia. The burden of anemia among lactating mothers in Ethiopia was higher than those who were neither pregnant nor breastfeeding. To date, there is limited evidence on spatiotemporal patterns of anemia among lactating mothers in the country. Exploring the spatial patterns of maternal anemia is vital to design and monitor effective intervention programs. Therefore this study aimed to explore spatiotemporal patterns of anemia among lactating mothers in Ethiopia over the past one and half-decades.

### Methods

A total of 11,989 lactating mothers were included from the three consecutive Ethiopian Demographic and Health Surveys(2005, 2011, and 2016). The trend of anemia over the three surveys was showed. Furthermore, spatial autocorrelation analysis, cluster and outlier analysis, hotspot analysis, spatial interpolation, and spatial scan statistics were carried out to identify geographically risk areas of anemia among lactating mothers in Ethiopia. Finally, the most anemia risk areas were detected consistently by different spatial analytic methods in each survey.

### Results

Anemia during lactation had an increasing trend from 2011 to 2016 in all regions of Ethiopia. It was also spatially clustered over three survey periods (Moran's I: 0.102–0.256, P<0.01). The hotspot areas were detected in Afar, Somali, Gambela, Dire Dawa, and Oromia regions during the last fifteen years. In 2005 and 2011, a total of 100 most likely clusters

available data. Any researcher can access data after becoming an Authorized user. Once registered and access permission has been provided, users may download the datasets from the required countries free of charge. Therefore, all the data underlying the findings are freely available from www.measuredhs.com.

**Funding:** The author(s) received no specific funding for this work.

**Competing interests:** The authors have declared that no competing interests exist.

**Abbreviations:** CSA, Central Statistical Agency; EA, Enumeration Area; EDHS, Ethiopian Demographic and Health survey; SNNP, South Nation Nationality and Peoples Region; WHO, World Health Organization.

(Loglikelihood Ratio(LLR) = 8.8, P<0.05, and LLR = 45.94, P<0.001, respectively) were identified in the Afar region. However, in the 2016 survey period, primary clusters were shifted to the Somali region where 57 clusters (LLR = 72.73, P<0.001) were detected in the entire region. Besides, the risk prediction map showed that the eastern part of the country was at a higher risk of anemia during lactation.

## Conclusion

Anemia during lactation was spatially clustered in Ethiopia. High-risk areas were detected in the eastern part of Ethiopia prominently in the Afar and Somali regions. Therefore, public health intervention activities designed in a targeted approach to impact high-risk populations in those hot spot areas wound be helpful to reduce anemia in Ethiopia.

## Introduction

Anemia is a condition with a low hemoglobin level with a cutoff point <110 g/L for pregnant women and <120 g/L for non-pregnant women [1]. It is characterized by a decreased number of red blood cells or hemoglobin level that results in the insufficient oxygen-carrying capacity of blood to meet the cellular metabolic demand of the body. Anemia can occur due to nutritional deficiencies (the most common cause), acute and chronic inflammations, parasitic infections, and acquired or inherited disorders that affect the synthesis of hemoglobin and production or survival of red blood cells [2, 3].

Anemia is a worldwide public health problem affecting both developing and developed countries that occurs in all population groups of the human being [2]. Globally, 38% of pregnant women and 29% of non-pregnant women were anemic in 2011.Of these, pregnant women in low-income and middle-income countries (LMICs)had high rates of anemia, in which the highest prevalence rates are reported in Central and West Africa (56%), South Asia (52%) and East Africa (36%) [4]. Similarly, a large proportion of non-pregnant women were reportedly anemic in West and Central Africa (48%), South Asia (47%), and East Africa (28%) [4]. In Ethiopia, anemia prevalence among reproductive-age women declined from 27% in 2005 [5] to 17% in 2011 [6] but it has increased to 24% in 2016 [7]. The prevalence rate of anemia among women who are lactating in Ethiopia was 29% which was higher than those who are neither pregnant nor lactating (21%) [7]. Furthermore, different studies had shown the burden of anemia in Ethiopia varies across different geographic locations [8–17]. This might be because of the existence of diverse contextual and geographically variable factors like diet and the incidence of infectious diseases [18, 19].

Anemia is associated with poor social and economic development, an increased risk of child mortality [2, 3], maternal mortality [20], depression [21, 22], raised blood pressure [23, 24], low birth weight, and preterm birth [25]. This makes anemia to be one of the global health priority areas, especially in resource-limited areas [18]. Therefore reducing anemia is considered as an essential part of improving the health of women, and the WHO has set a global target of achieving a 50% reduction of anemia among women of reproductive age by 2025 [26].

Lactating mothers are vulnerable to anemia because of maternal iron depletion during lactation as well as blood loss during childbirth [27]. Studies have indicated that, even though breast milk is not a good source of iron, the quality of breast milk is maintained at the expense of maternal stores [28]. The postpartum anemia was highest in mothers who are anemic

during their gestational period [29]. Moreover, lactating mothers are highly susceptible to iron depletion if the energy and nutrient intake in their diets is inadequate. Besides, the lactating mothers begin the postnatal period after having iron-depleted through the continuum from pregnancy to childbearing [30].

Though there are studies conducted on the determinants of anemia among lactating mothers [31–33] in Ethiopia, to date, the risk areas (hot spot) of anemia among lactating mothers are not identified. Thus, this study aimed to explore the spatiotemporal patterns of anemia among lactating mothers in Ethiopia over the last one and half-decades to point out whether there was either the shift or improvement in anemia risk areas following intervention programs in between the survey periods in Ethiopia. Therefore, detecting the geographic variation of anemia during lactation is important to prioritize and design targeted intervention programs to reduce anemia especially in those areas with a consistently higher risk of anemia over time. Besides, since the burden of anemia has been used as a measurable indicator of soil-transmitted helminthiasis [34], understanding the geographical distribution of anemia can help to target prevention and control mechanisms for these parasitic infections in the area.

## Methods

### Study design and setting

An in-depth analysis of data from Ethiopian Demographic and Health Surveys (EDHS) 2005, 2011, and 2016 was undertaken for this study. Ethiopia ($3^o$ -$14^o$ N and $33^o$ - 48˚E) is located in the horn of Africa (Fig 1). The country covers 1.1 million Sq. km and has a great geographical diversity, which ranges 4550 m above sea level down to the Afar depression to 110 m below sea level. There are nine regional states(Afar, Amhara, south nation nationality and peoples, Benshangul Gumuz, Gambela, Harari, Oromia, Somalia, and Tigry) and two city administrations (Addis Ababa and Dire Dawa). These regions were again subdivided into 68 zones, 817 districts, and 16,253 kebeles (lowest local administrative units of the country) in the administrative structure of Ethiopia [7].

### Data source and sampling

The Demographic and Health Survey (DHS) Program provides publicly free access to survey data for responsible researchers. Therefore, we accessed the datasets using the website www.measuredhs.com after the reasonable request of the Demographic and Health Survey(DHS). Researchers can access the data free of charge and can replicate our study findings in their entirety by directly obtaining the data. The detailed description of each dataset and other relevant information could be obtained elsewhere [35].

Accordingly, the Ethiopian Demographic and Health Survey (EDHS) was used for the current study which has collected data on national representative samples of all populations including reproductive-age (15–49) women every five years interval. To date, four surveys had been conducted and anemia was included as a key indicator since the 2005 survey.

In 2005 and 2011 surveys, 540 (139 urban and 401 rural areas) and 624 (187 urban and 437 rural) enumeration areas (EAs) were selected using systematic random sampling with probability proportional to size. A total of 14,645 households (17,817 eligible reproductive age women), and 14,645 households (14,717 eligible reproductive age women), respectively, were included. In 2016 EDHS, 645 EAs (202 urban and 443 rural) were selected. Of these, 18008 households and 16,583 eligible reproductive-age women were included. In the current study, a total of 11,989 (weighted) lactating mothers were included from the three surveys (Table 1). Besides, geographic coordinate data (latitude and longitude coordinates) were also taken from selected enumeration areas in all three surveys. These geo-referenced data were accessed

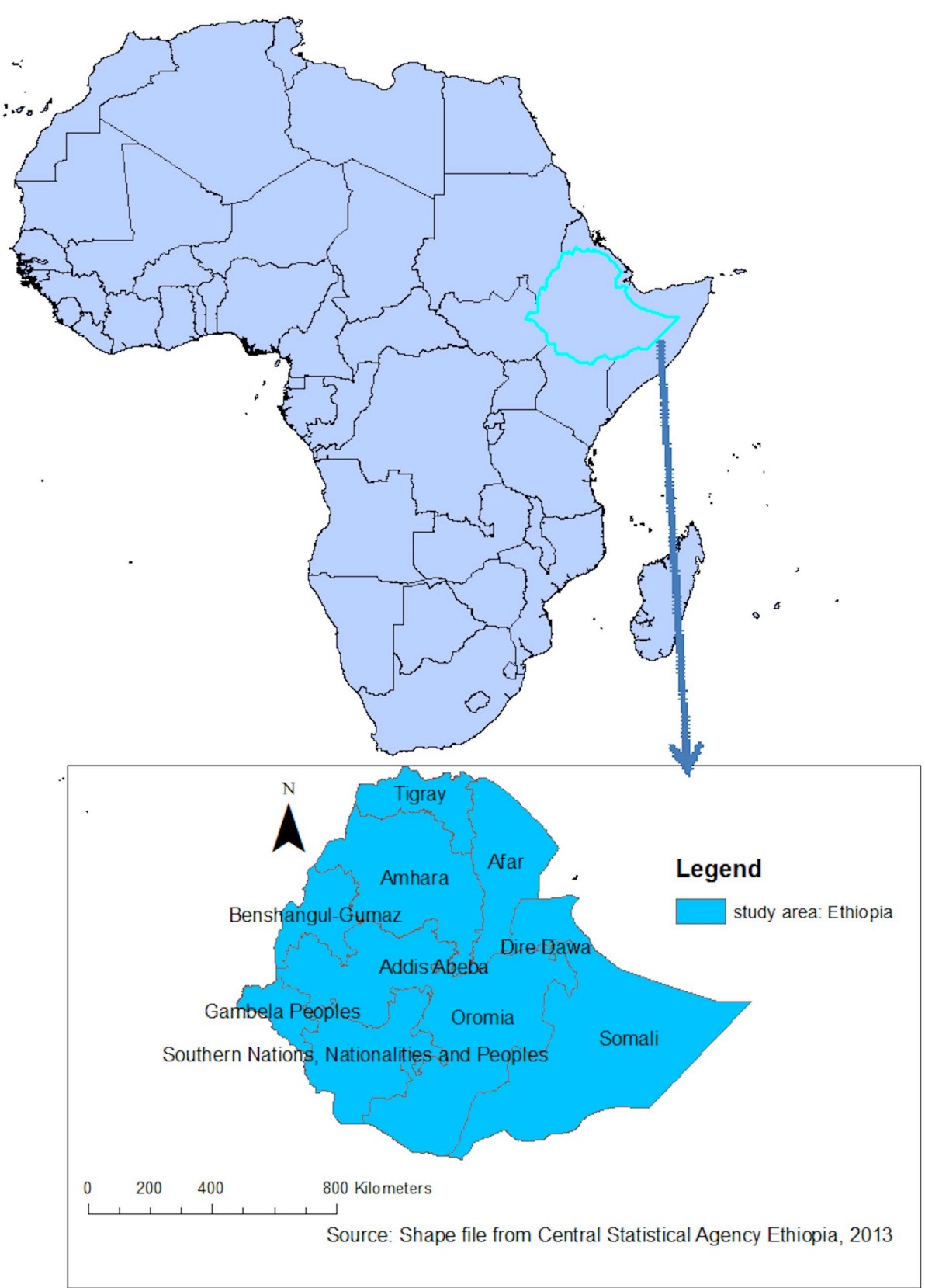

**Fig 1. Map of Ethiopia where the three surveys were undertaken with nine regions and two administrative cities (Source: Shapefile from Central Statistical Agency, Ethiopia, 2013).**

**Table 1. Total number of lactating mothers included in 2005, 2011 and 2016 EDHS by region, Ethiopia.**

| Regions | Year of Survey | | | |
|---|---|---|---|---|
| | 2005 | 2011 | 2016 | Total |
| Tigry | 197 | 446 | 449 | 1092 |
| Afar | 116 | 351 | 332 | 799 |
| Amhara | 327 | 659 | 518 | 1504 |
| Oromia | 365 | 721 | 621 | 1707 |
| Somali | 186 | 333 | 369 | 888 |
| Benshangul Gumuz | 164 | 438 | 364 | 966 |
| SNNP | 398 | 689 | 556 | 1643 |
| Gambela | 133 | 393 | 392 | 918 |
| Hareri | 130 | 336 | 287 | 753 |
| Addiss Ababa | 162 | 350 | 379 | 891 |
| Dire Dawa | 107 | 331 | 390 | 828 |
| Total | 2,285 | 5,047 | 4,657 | 11,989 |

SNNP: South Nation Nationality and Peoples Region.

through the web page of the international DHS Program after justifying the purpose of the study.

## Outcome variable

In the three Ethiopian Demographic and Health Surveys, the hemoglobin level was measured for those eligible reproductive age mothers after having consent and it was adjusted for altitude [5–7]. This altitude adjusted hemoglobin was available in the dataset for each survey. Therefore, the current study was based on the altitude adjusted hemoglobin level which was already provided in the EDHS data. After the categorization of the hemoglobin level, lactating mothers with a hemoglobin level <120 g/L were considered as anemic and otherwise nonanemic. Finally, the weighted proportion of anemia per cluster was used for further spatial analysis.

## Spatial analysis

The spatial autocorrelation (Global Moran's I) statistic was used to evaluate whether the anemia patterns are dispersed, clustered, or randomly distributed during the three survey periods in Ethiopia. The decision was made based on the calculated Moran's I values. When the Moran'I value is close to−1 indicates anemia is dispersed, whereas Moran's I close to +1 indicates anemia is clustered in the study area. However, the Moran's I value zero shows a random distribution of anemia. Once it was confirmed that the global distribution of anemia is nonrandom, the local Moran's I was used to investigate the local level cluster locations of anemia in Ethiopia. Local Moran's I identify hotspot clusters (High-High), and cold spot clusters (Low-Low). It also measures outliers where high values were surrounded primarily by low values (High-Low), and outliers in which low values were surrounded primarily by high values (Low-High) [36, 37]. This spatial analysis technique was employed to detect the local level risk areas of anemia and its outliers on a separate map.

In addition to local Moran's I, Gettis-OrdGi* statistics was computed to measure how spatial autocorrelation of anemia among lactating mothers varies across different locations in Ethiopia. Hotspot analysis computes Z-score and p-value to determine the statistical significance of the clustering of anemia over the study area at different significance levels simultaneously [38]. In this analysis, the p-value associated with a 95%, 90%, and 99% confidence

level would have provided to decide the existence of significant clustering. Areas at high risk (hotspot) of lactational anemia (the statistical output with high Gi*) and areas at low risk (cold spot) of anemia during lactation (the statistical output with low Gi*) were detected [36, 37, 39].

The spatial interpolation technique was applied to predict the unsampled areas from sampled measurements [40]. Ordinary Kriging spatial interpolation method was used to predict raster surface from point data. Therefore, smooth surfaces for the risk areas of anemia among lactating mothers was indicated on the anemia risk map.

Identifying most likely clusters was done using the spatial Scan statistical method, a method which is widely recommended as it is very important in detecting local clusters and has higher power than other available spatial statistical methods [41]. Therefore, spatial scan statistical analysis was employed to test for the presence of statistically significant spatial clusters of anemia using Kuldorff'sSaTScan version 9.4 software [42]. The spatial scan statistic uses a scanning window that moves across the study area. Women with anemia were taken as cases and non-anemic ones were considered as controls to fit the Bernoulli model. The default maximum spatial cluster size of <50% of the population was used as an upper limit, which allowed both small and large clusters to be detected and ignored clusters that contained more than the maximum limit. For each potential cluster, a likelihood ratio test statistic was used to determine if the number of observed anemia cases within the potential cluster was significantly higher than expected or not. The primary and secondary clusters are identified and ranked based on their likelihood ratio based on 999 Monte Carlo replications. Therefore most likely risk areas of anemia among lactating mothers in three consecutive surveys were indicated in consecutive spatial maps.

### Ethical consideration

Ethical clearance was approved by an Institutional Ethical Review Committee of the Institute of Public Health, College of Medicine and Health Sciences, University of Gondar. The approval letter for the use of the EDHS data set was also gained from the Measure DHS (ORC MACRO). No information obtained from the data set was disclosed to any third person.

## Results

### Trends and spatial distribution of anemia among lactating mothers

Even though it had decreased from 2005 to 2011 almost in all regions, its prevalence increased from 2011 to 2016 in all regions including the two administrative cities. The highest prevalence was observed in the Somali regional state (68%) and the Afar region (47%) in 2016 (Fig 2).

Moreover, the exploratory visualization of the spatial distribution of anemia showed a wide geographic variation across regions in three surveys. The highest proportions were observed in Pastoral regions in all three surveys which were consistent with the observed trend (Fig 3A– 3C).

### Spatial autocorrelation of anemia among lactating mothers

The spatial patterns of anemia among lactating mothers were found to be non-random during the three study periods EDHS (Fig 4A–4C). The Global Moran's I values ranged from 0.101 to 0.26, indicating that there was significant clustering of anemia among lactating mothers in the country. The clustering pattern in all three study periods was highly significant (>90%) (Fig 4A–4C, Table 2).

The clustered patterns (on the right sides) show high rates of anemia occurred over the study area. The outputs have automatically generated keys on the right and left sides of each

**Fig 2. Trends of anemia among lactating mothers in Ethiopia overtime across regions 2005, 2011, and 2016.**

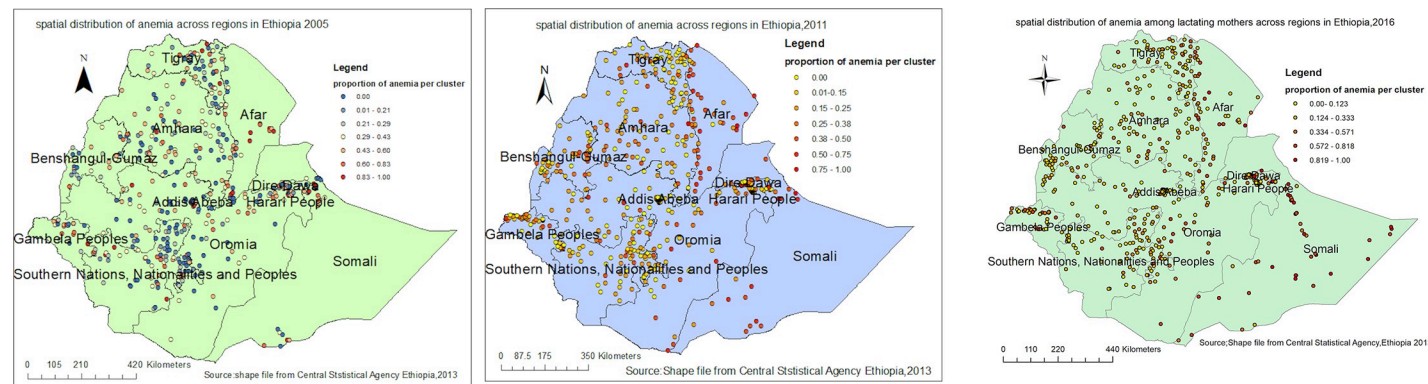

**Fig 3.** a-c: Spatial distribution of anemia among lactating mothers in Ethiopia 2005 (a), 2011 (b), 2016 (c) EDHS (Source: Shapefile from Central Statistical Agency, Ethiopia, 2013).

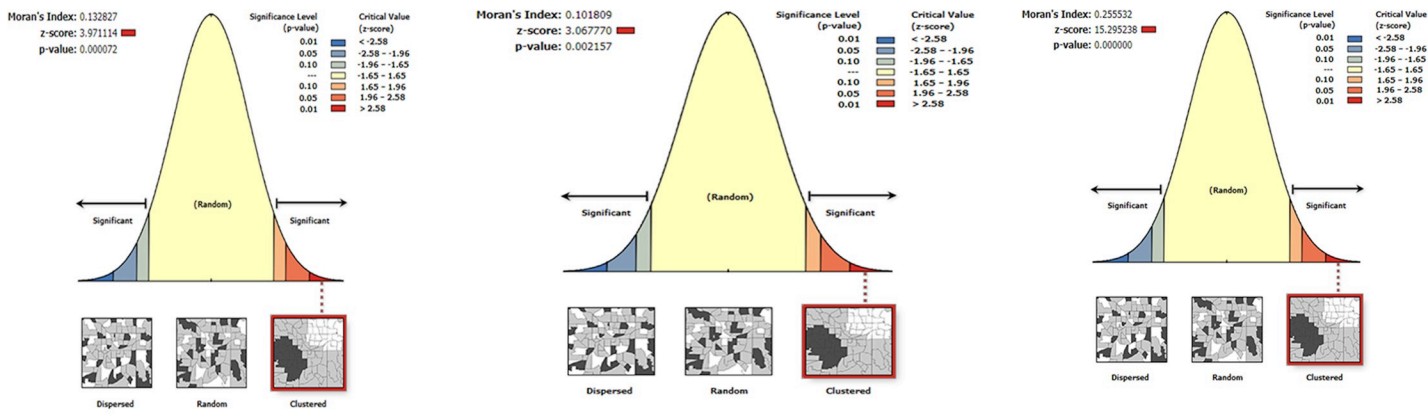

**Fig 4.** a-c: Spatial autocorrelation of anemia among lactating mothers in Ethiopia, 2005 (a), 2011 (b), 2016 (c) (Source: Shapefile from Central Statistical Agency, Ethiopia, 2013).

panel. Auto-generated interpretations available underneath each figure show that the likelihood of clustered patterns occurred by random chance is less than 1%.

## Spatial epidemiology of anemia among lactating mothers

Both Figs 5 and 6 indicate the consistently similar geographical distribution of risk areas of anemia among lactating mothers during the three survey periods. Each spot on the map indicates a single enumeration area. The hotspot (enumeration areas with high anemia risk) areas were found in Afar (in all surveys); Somali, Dire Dawa, Harari(in 2011 and 2016 surveys); and Gambella (2005 and 2016 surveys) regions. Whereas, Addis Ababa, Oromia, Amhara, Tigray, SNNP, Benshangul in all surveys; and Gambela in 2011 survey were identified as cold spot (enumeration areas with low anemia risk) regions (Figs 5 and 6). The outliers were found in Addis Ababa, Dire Dawa, Oromia, SNNP, and southern parts of Afar in all surveys; Southwest Amhara and Benshangul-Gumuz in 2016 survey; and Northern part of Gambela in 2005 survey (Fig 5A–5C).

## Spatial interpolation

In the 2005 survey (Fig 7A), the Afar (Eastern part),Somali (west), and Gambela regions were predicted as a more risky area of anemia during lactation as compared to other regions. Whereas, in the 2011 survey (Fig 7B), the entire Afar, the northern part of Dire Dawa,Somali and eastern border of Oromia regions were identified as risk areas. The predicted risk of anemia during lactation was almost shifted to the entire Somali region in the 2016 survey (Fig 7C).

**Table 2. Spatial autocorrelation analysis of anemia among lactating mothers in Ethiopia, 2005, 2011, and 2016.**

| EDHS study periods | Observed moran's I | Expected moran's I | Z-score | P-value |
|---|---|---|---|---|
| 2005 | 0.13* | -0.02 | 3.97 | <0.01* |
| 2011 | 0.10* | -0.01 | 3.07 | <0.05* |
| 2016 | 0.26* | -0.01 | 15.30 | <0.01* |

*The observed Moan's I value is greater than the expected value and the p-value< 0.05, which revealed that the spatial dependency of anemia is statistically significant during three periods.

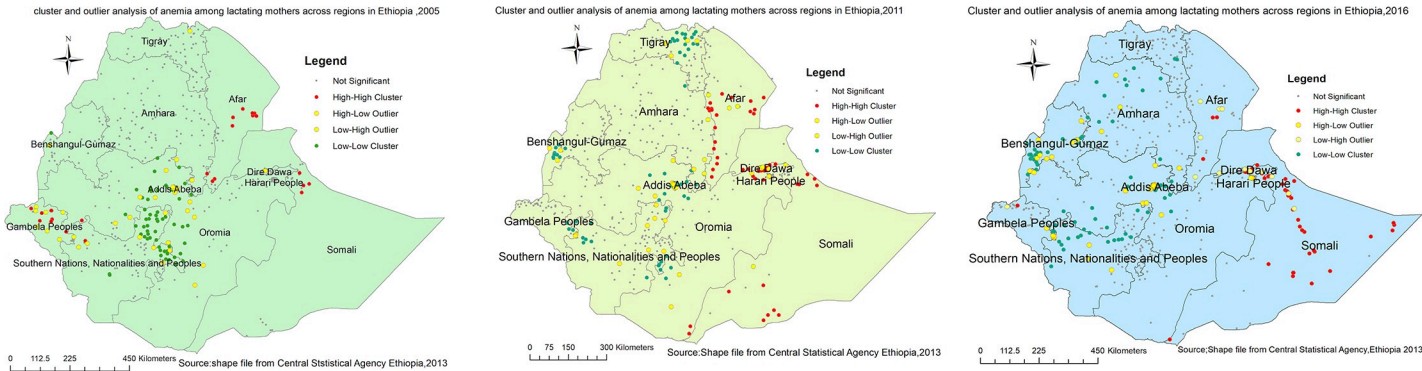

**Fig 5.** a-c: Cluster outlier identification ofanemia among lactating mothers in Ethiopia, 2005 (a), 2011 (b), 2016 (c) EDHS (Source: Shapefile from Central Statistical Agency, Ethiopia, 2013).

### Spatial scan statistical analysis

Overall, a total of 157 most likely (primary) clusters were detected across three EDHS surveys. Of these, 8 significant primary clusters were identified in 2005. The spatial scanning window for these clusters was located in the eastern part of the Afar region and border areas of Dire Dawa. It was centered at 11.559215 N, 41.505535 E with a radius of 94.66 km, a relative risk (RR) of 2.65, and a Log-likelihood Ratio (LLR) = 8.80 at p-value<0.05) (Table 3, Fig 8A). The lactating women within the spatial window had 2.65 times higher risk of being anemic as compared to women outside the spatial window.

Where as, in 2011, spatial scan statistics detected a total of 92 primary clusters. The spatial window for these clusters was located in the almost entire Afar, border areas of Southern Tigray and South-Eastern part of Amhara regions. It was centered at 11.494762 N, 41.566624 E with a radius of 229.79 km, RR of 1.98, LLR = 45.94, p-value<0.001 (Table 3, Fig 8B). It showed that lactating women within the spatial window had 2 times higher risk of anemia as compared to women outside the spatial window.

Furthermore, a total of 57 primary clusters were identified in the 2016 survey. The spatial window for these clusters was located in the entire Somali, and eastern border areas of Dire Dawa and Oromia regions, centered at 6.023458 N, 44.807507 E) with a radius of 462.80 km, RR of 2.27, LLR of 72.73 at p-value<0.001 (Table 3, Fig 8C). Lactating women within the spatial window had 2.27 times higher risk of anemia as compared to women outside the spatial window (Table 3, Fig 8C)s.

### Discussion

The findings of this study showed that anemia during lactation was non-random at the national and regional levels. Significant clusters were consistently detected in the Afar region during all surveys (2005, 2011 and 2016). A total of 100 (8 in 2005; 92 in 2011) primary clusters were identified in this region. However, in 2016,57 significant primary clusters were shifted to the Somali region. Moreover, anemia risk prediction map showed that the eastern part of the country to be at a higher risk of anemia during lactation in all three surveys. The geographical difference of anemia across the regional states might be attributable to the regional variation of food consumption preferences [43, 44] and differences in availability of healthcare facilities [45]. This study revealed that the regions which were less developed [46] as compared to other Ethiopian states were at high risk of anemia. The possible explanation could be a lack of clean water and unimproved latrine facilities which would increase the occurrence of soil-

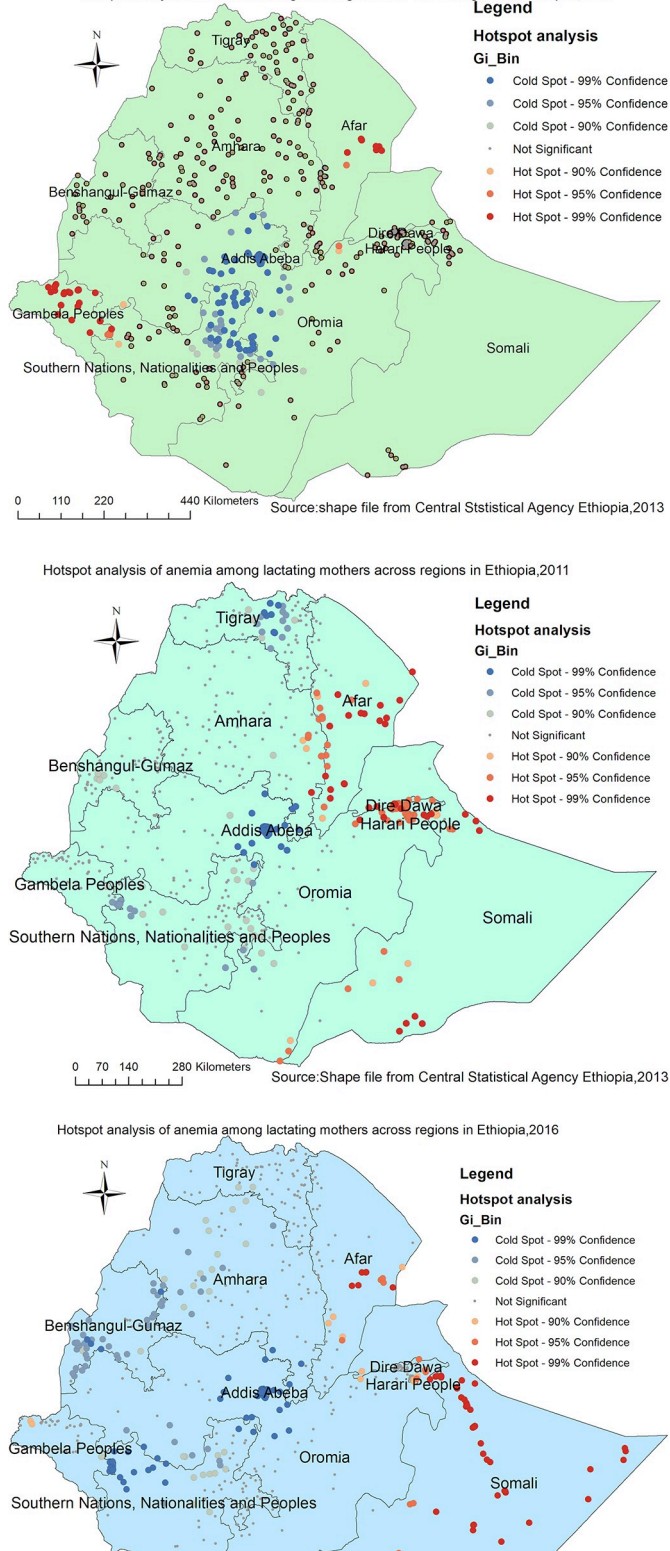

**Fig 6.** a-c: Hot spot identification of anemia among lactating mothers in Ethiopia 2005 (a), 2011 (b), 2016 (c) EDHS (Source: Shapefile from Central Statistical Agency, Ethiopia, 2013).

ordinary kriging  interpolation of anemia among lactating mothers across regions in Ethiopia ,2005

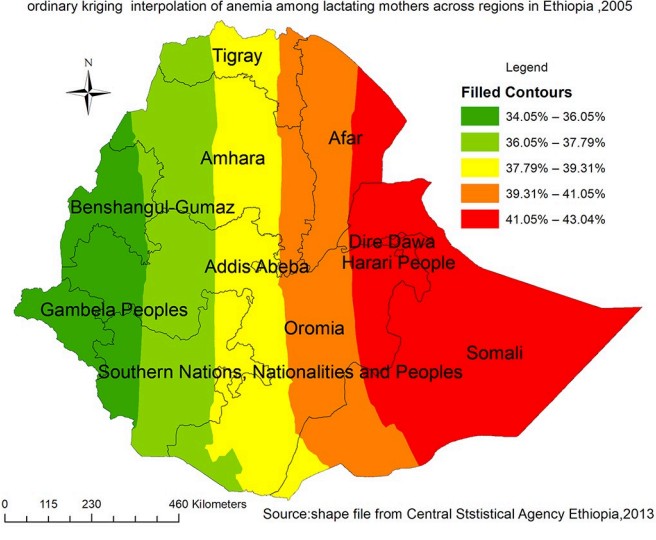

Ordinary kriging interpolation of anemia among lactating mothers across regions in Ethiopia,2011

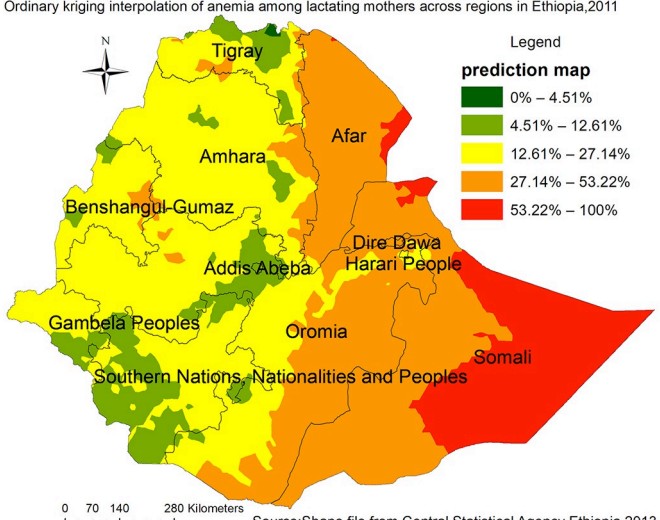

ordinary kriging  interpolation  of anemia among lactating mothers across regions in Ethiopia,2016

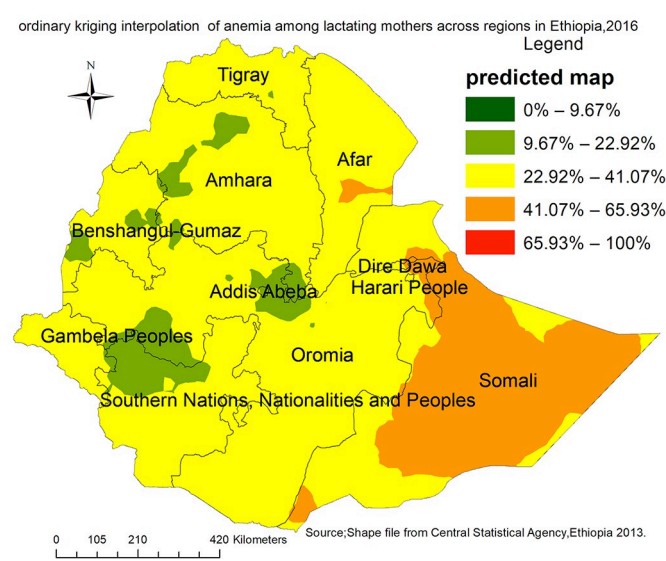

**Fig 7.** a-c: Spatiotemporal interpolation of anemia among lactating mothers in Ethiopia 2005 (a), 2011 (b), 2016 (c) (Source: Shapefile from Central Statistical Agency, Ethiopia, 2013).

transmitted infections [47] that might in turn lead to anemia [48]. Besides, the observed geographical variation in anemia risk could be due to the incidence of communicable and non-communicable diseases [49].

Despite several maternal health interventions in Ethiopia, the prevalence of anemia among lactating mothers had shown an increasing trend in all regions from 2011 to 2016 (Fig 2). The largest increase was observed in the Somali and Afar regions. Besides, the spatial autocorrelation analysis result indicated that anemia had a spatial dependency in 2005, 2011, and 2016. This spatial heterogeneity of anemia clustering was again observed prominently in Afar and Somali regions. Thus, the spatial clustering of anemia was more or less consistently higher in the Afar region in all three surveys and the Somali region in the latest two surveys (2011 and 2016). These could be due to the inaccessibility of health services, shortage of safe and adequate drinking water supply, and endemicity of malaria in these areas as compared to other regions of Ethiopia [50]. Though iron supplementation is an essential service for prevention of anemia among lactating mothers [2], its coverage is very low in these pastoral regions (Afar and Somali) which might be responsible for a high prevalence and hotspot areas of anemia in these regions as compared to other parts of Ethiopia [7, 51]. Besides, the compliance of lactating mothers to the use of iron supplementation service in these regions is below the recommended level [52] which might lead to an increased risk of anemia. The other possible reason behind the consistent hotspot areas in these regions could be nutritional problems such as lack of dietary diversity, consumption of camel, and cow milk which had relatively low iron content and the seasonal variation of food they consume (pastoral community) [53].

The findings of this study have valuable policy implications for health program design and interventions. The anemia hotspot areas can be easily identified even at the district level to take local interventions. It may also be helpful to give priority for regions that were consistently at higher risk of anemia over time. In general, these findings are supremely important for the Ministry of Health, Health Bureaus, and partners to develop intervention programs against anemia.

**Table 3. Significant spatial clusters of anemia among lactating mothers in Ethiopia, EDHS 2005, 2011, 2016.**

| Years | Clusters | Enumeration areas (clusters) detected | Coordinate/radius | Population | Cases | RR | LLR | P-value |
|---|---|---|---|---|---|---|---|---|
| 2005 | 1* | 186,248,227,515,59,413,47,18 | 11.56 N, 41.50 E) / 94.66 km | 16 | 13 | 2.65 | 8.80 | <0.05 |
| 2011 | 1* | 314, 215, 62, 589, 164, 600, 512, 397, 106, 579, 68, 65, 604, 392, 617, 296, 210, 84, 445, 33, 99,414, 643, 577, 191, 79, 85, 67, 26, 231, 478, 293, 423, 433, 549, 133, 44, 323, 596, 572, 366, 59, 590, 616, 529, 464, 159, 581, 205, 629, 501, 110, 376, 645, 308, 194, 400, 182, 203, 170, 352, 455, 285, 448, 446, 499, 602, 345, 371, 420, 421, 386, 138, 560, 270, 95, 286, 51, 236, 436, 200, 66 | (11.49 N, 41.57E) / 229.79 km | 528 | 210 | 1.98 | 45.9 | <0.01 |
| 2016 | 1* | 562, 213, 619, 123, 524, 438, 261,46, 138, 92, 490, 543, 492, 85, 358, 164, 77, 171, 198, 629, 95, 497, 278, 521, 588, 458, 553, 269, 318, 378, 187, 630, 214, 251, 573, 556, 239, 116, 22, 520, 33, 568, 277, 480, 527, 208, 64, 439, 57, 8, 210, 186, 394, 454, 436, 566, 212 | (6.02 N, 44.81 E) / 462.80 km | 277 | 181 | 2.27 | 72.73 | <0.01 |

*primary clusters.

spatial scan statistics of anemia among lactating mothers across regions in Ethiopia ,2005

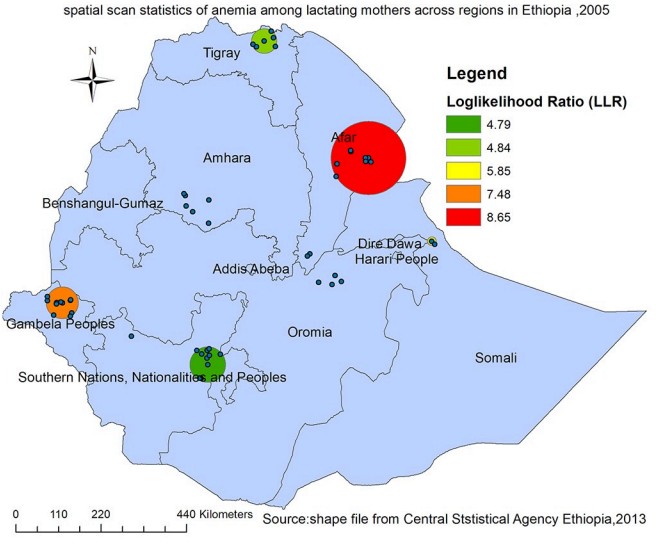

spatial scan statistics of anemia among lactating mothers across regions in Ethiopia,2011

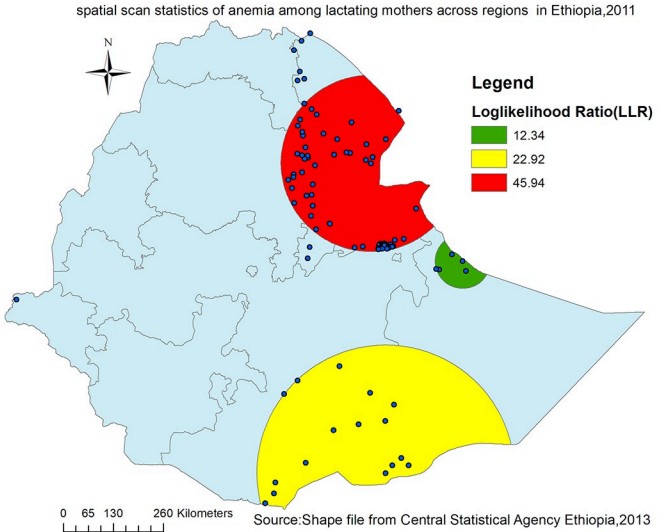

spatial scan statistics of anemia among lactating mothers across regions in Ethiopia, 2016

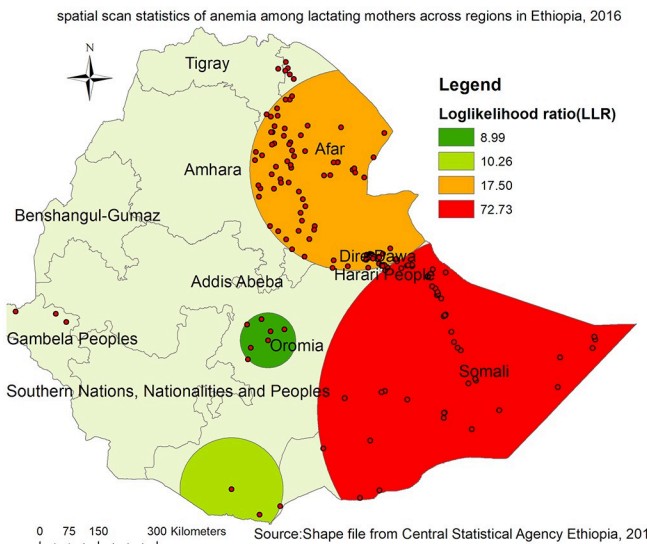

**Fig 8.** a-c. spatiotemporal patterns of primary clusters of anemia among lactating mothers in Ethiopia 2005 (a), 2011 (b), and 2016 (c) (Source: Shape file from Central Statistical Agency, Ethiopia, 2013).

## Strength and weakness of the study

This study had several strengths. First, the study was based on nationally representative large datasets, and thus it had adequate statistical power. Second, the estimates of the study were done after the data were weighted for the probability sampling and non-response, to make it representative at national and regional levels. Third, the use of GIS and SaTScan statistical softwares helped to detect statistically significant hotspot areas of anemia across the surveys consistently that will help to design effective public health intervention programs. However, our study is not free from limitations. First, the location data values were shifted 1–2 km for urban and 5 km for rural areas for data confidentiality issues. Consequently, this may lead to a challenge to know the exact location of cases. Second, spatial modeling was not conducted to identify the spatial determinates in those risk areas.

## Conclusion

Though a declining pattern of anemia among lactating mothers was observed from 2005 to 2011, it increases from 2011 to 2016 in almost all regions with a higher prevalence in Afar and Somali regions. Besides, it was spatially clustered across regions in Ethiopia. The most prominent risk areas of anemia were again detected in Afar and Somali regions more or less consistently overtime in the last one and half-decade. Therefore, public health intervention activities designed in a targeted approach to impact high-risk populations as well as geographic regions were vital to reduce anemia among lactating mothers in Ethiopia.

## Acknowledgments

The authors would like to thank MEASURE DHS for their permission to access the DHS dataset.

## Author Contributions

**Conceptualization:** Alemneh Mekuriaw Liyew, Sewnet Adem Kebede, Chilot Desta Agegnehu, Yigizie Yeshaw, Getayeneh Antehunegn Tesema.

**Data curation:** Alemneh Mekuriaw Liyew, Sewnet Adem Kebede, Chilot Desta Agegnehu, Getayeneh Antehunegn Tesema.

**Formal analysis:** Alemneh Mekuriaw Liyew, Sewnet Adem Kebede, Achamyeleh Birhanu Teshale, Adugnaw Zeleke Alem, Yigizie Yeshaw, Getayeneh Antehunegn Tesema.

**Investigation:** Alemneh Mekuriaw Liyew, Sewnet Adem Kebede, Adugnaw Zeleke Alem, Yigizie Yeshaw.

**Methodology:** Alemneh Mekuriaw Liyew, Sewnet Adem Kebede, Chilot Desta Agegnehu, Adugnaw Zeleke Alem, Yigizie Yeshaw, Getayeneh Antehunegn Tesema.

**Software:** Alemneh Mekuriaw Liyew, Sewnet Adem Kebede, Chilot Desta Agegnehu, Adugnaw Zeleke Alem, Yigizie Yeshaw.

**Supervision:** Alemneh Mekuriaw Liyew, Adugnaw Zeleke Alem, Yigizie Yeshaw, Getayeneh Antehunegn Tesema.

**Validation:** Alemneh Mekuriaw Liyew, Sewnet Adem Kebede, Chilot Desta Agegnehu, Achamyeleh Birhanu Teshale, Yigizie Yeshaw.

**Visualization:** Alemneh Mekuriaw Liyew, Sewnet Adem Kebede, Chilot Desta Agegnehu, Yigizie Yeshaw.

**Writing – original draft:** Alemneh Mekuriaw Liyew, Sewnet Adem Kebede, Achamyeleh Birhanu Teshale, Adugnaw Zeleke Alem, Getayeneh Antehunegn Tesema.

**Writing – review & editing:** Alemneh Mekuriaw Liyew, Sewnet Adem Kebede, Achamyeleh Birhanu Teshale, Adugnaw Zeleke Alem, Yigizie Yeshaw, Getayeneh Antehunegn Tesema.

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
