## [Decision Letter · Decision Letter 0]

15 Jan 2020

PONE-D-19-30322

Spatial patterns of anemia among lactating mothers in Ethiopia: Data from Ethiopian Demographic and Health Surveys (2005, 2011 and 2016)

PLOS ONE

Dear Mr liyew,

Thank you for submitting your manuscript to PLOS ONE. After careful consideration, we feel that it has merit but does not fully meet PLOS ONE’s publication criteria as it currently stands. Therefore, we invite you to submit a revised version of the manuscript that addresses the points raised during the review process.

We would appreciate receiving your revised manuscript by Feb 29 2020 11:59PM. To enhance the reproducibility of your results, we recommend that if applicable you deposit your laboratory protocols in protocols.io, where a protocol can be assigned its own identifier (DOI) such that it can be cited independently in the future. For instructions see: http://journals.plos.org/plosone/s/submission-guidelines#loc-laboratory-protocols

We look forward to receiving your revised manuscript.

Kind regards,

William Joe

Academic Editor

PLOS ONE

https://www.ncbi.nlm.nih.gov/pmc/articles/PMC5472978/

https://www.ncbi.nlm.nih.gov/pubmed/30948614

In your revision ensure you cite all your sources (including your own works), and quote or rephrase any duplicated text outside the methods section. Further consideration is dependent on these concerns being addressed.

3. Please ensure you have thoroughly discussed any potential limitations of this study within the Discussion section, for example, the potential impact of confounding variables.

4. We note that Figures 1, 3, 5, 6, 7 and 8 in your submission contain [map/satellite] images which may be copyrighted. All PLOS content is published under the Creative Commons Attribution License (CC BY 4.0), which means that the manuscript, images, and Supporting Information files will be freely available online, and any third party is permitted to access, download, copy, distribute, and use these materials in any way, even commercially, with proper attribution. For these reasons, we cannot publish previously copyrighted maps or satellite images created using proprietary data, such as Google software (Google Maps, Street View, and Earth). For more information, see our copyright guidelines: http://journals.plos.org/plosone/s/licenses-and-copyright.

a)    You may seek permission from the original copyright holder of Figures 1, 3, 5, 6, 7 and 8 to publish the content specifically under the CC BY 4.0 license.  

Reviewers' comments:

Reviewer's Responses to Questions

**Comments to the Author**

1. Is the manuscript technically sound, and do the data support the conclusions?

Reviewer #1: No

2. Has the statistical analysis been performed appropriately and rigorously? 

Reviewer #1: No

3. Have the authors made all data underlying the findings in their manuscript fully available?

Reviewer #1: No

4. Is the manuscript presented in an intelligible fashion and written in standard English?

Reviewer #1: No

5. Review Comments to the Author

Reviewer #1: Reviewer’s Report

Spatial patterns of anemia among lactating mothers in Ethiopia: Data from Ethiopian Demographic and Health Surveys (2005, 2011 and 2016)

Anemia among human population is a major public health problem that affects populations in both rich and poor countries. Its primary cause is iron deficiency (IDA) and other various biological and non-biological factors. Anaemia is a common health problem in Ethiopia also where the latest DHS survey found around 24 percent women of reproductive age (WRA) were anaemic, ranges from a low of around only 16 percent to a high of close to 60 percent in different parts of Ethiopia. In the context, the spatial exploration of anaemia especially among lactating women is also an important issue and the authors have tried to study the concentration and clustering of one of the major health problems but the authors have missed somewhere and needs further improvement in the manuscripts. The constructive comments and suggestions are as follows-

1. Introduction section has started in right way with definition and cause of anaemia with most related to biological factors but missing non-biological factors. Next, anaemia as a public health problem has discuses in brief flowed by global scenario of Anaemia. Then adverse effect of maternal anaemia has been discussed before moving on scenario of anaemia in Ethiopia. Introduction section ends with reason for selecting lactating women for the study purpose.

2. Here, some important aspects found missing. Like, previous studies pertain to anaemia among WRA in Ethiopia, evidences and studies related to spatial pattern of anaemia among WRA in the study country. Reason for considering lactating women for study needs to be strengthened. Authors are also advised to do review of literature extensively as many relevant studies is available on anaemia among WRA in Ethiopia conducted on same data source at different point of time. Some studies are very recent with latest round of country DHS data.

3. Data source, sampling design and setting needs to be rewrite in appropriate manner. Whereas measurements should be different sections and can be written accordingly. It would have been better if researchers would have provided region wise sample over the three study period.

4. As the author mentioned “The haemoglobin level was measured for those eligible mothers after having consent and it was adjusted for altitude”. This needs to be clarified as DHS data for the country provide measured haemoglobin level and the adjusted haemoglobin for altitude. So, which one indicator authors have used in the study. The cut off used in this study has not been mentioned in the methodology section.

5. Spatial analysis section starts with autocorrelation analysis to spatial scan statistical analysis. Here, the concern is that in many section, authors have provided only the theoretical notions of such the method but missed to elaborate in accordance and need of the undertaken study. Like, how SAC has been executed in the study has found missing. Second, section on SAC mentioned somewhere “leads to rejection of null hypothesis and indicates the presence of spatial correlation”. Does the author have posed research question in same way?

6. Hot spot analysis (HSA) using Gettis-OrdGi* statistics is a appropriate way to check the variation in spatiality across the study areas but missed to the process undertaken in the study. It would be better if authors had been reviewed available study for spatial analysis of anaemia in Ethiopia by Kelemu T.K, et (2019) study using latest wave of Ethiopia DHS data. So, author is advised to go through the paper and restructure the section appropriately.

7. Overall methodology section needs to tightened and strengthened.

8. Results are very poorly interpreted, seems researchers are in hurry. So, results sections had to be written in well manner. Figures 3 can be interpreted region wise and changes over period would be better idea. Interpretation in figure heading is not the academic practices, so author is suggested to follow the rigour and pattern/outline to prepare the manuscript.

9. It would have been better to elaborate the section on emerging hotspot areas based on the data available for the areas before doing interpolation analysis to detect the primary and secondary clusters.

10. In spatial scan analysis, authors have tires to provide glimpse of significant clusters separately for each survey period but it they can provide reason wise prominent clusters, it would be better for planning and programme intervention at the regional level within Ethiopia.

11. Table 3 does not have catchy and reliable approach. It would have been better if the researchers would have been mentioned detailed with name or within each of the cluster with maximum and minimum concentration of anaemic lactating mothers. Second, researcher can match for enumeration areas which were found common during the period of 2005, 2011 and 2016, cluster wise. The format of table 3 provided in the text can be produced as supplementary table or appendix table.

12. Discussion section needs to rewrite as it is just findings of study only in very limited way. It is important to discuss the findings of the study with other previous and existing studies on the issues.

13. In the conclusion section authors have mentioned “Findings suggest that giving priority attentions would be important on water, and other nutrition-related interventions on the identified hotspot areas to prevent and control anaemia incidence among lactating mothers”. The concern is that from where these findings emerged as it is not found anywhere and anyway in the manuscript.

14. Maps and figures are also not readable as are very hazy. So authors are also advised to rework for figure and maps to make it clear before submitting it the journal.

Overall, the paper needs to revise with appropriate framework for the country undertaken for the policy and programmatic point of view.

6. PLOS authors have the option to publish the peer review history of their article (what does this mean?). If published, this will include your full peer review and any attached files.

Reviewer #1: Yes: Rajesh Raushan

---

## [Author Response · Author response to Decision Letter 0]

24 Mar 2020

Rebuttal letter Date 1/27/2020

PONE-D-19-30322

Spatial Patterns of Anemia among Lactating Mothers in Ethiopia: Data from Ethiopian Demographic and Health Surveys (2005, 2011 and 2016)

Alemneh Mekuriaw Liyew

Plos One

Dear all,

We would like to thank for these constructive, building and improvable comments on this manuscript that would improve substance and content of the manuscript. We considered each comments and clarification questions of editors and reviewers on the manuscript thoroughly. Our point-by-point responses for each comment and questions are described in detailed on the following pages. Further, the details of changes were shown by track changes in the supplementary document attached. The manuscript language was checked by language professionals and we follow journal guideline. I have attached recent comments in point by point response.

Version 

Comments from reviewer

Comment 1: Introduction section has started in right way with definition and cause of anaemia with most related to biological factors but missing non-biological factors. Next, anaemia as a public health problem has discuses in brief flowed by global scenario of Anaemia. Then adverse effect of maternal anaemia has been discussed before moving on scenario of anaemia in Ethiopia. Introduction section ends with reason for selecting lactating women for the study purpose.

Response 1: Thanks in advance both editor and reviewer for your critical view to advance the quality of our manuscript. We improved accordingly. 

Comment 2: “Here, some important aspects found missing. Like, previous studies pertain to anaemia among WRA in Ethiopia, evidences and studies related to spatial pattern of anaemia among WRA in the study country. Reason for considering lactating women for study needs to be strengthened. Authors are also advised to do review of literature extensively as many relevant studies is available on anaemia among WRA in Ethiopia conducted on same data source at different point of time. Some studies are very recent with latest round of country DHS data.”

 Response 2: Thanks in advance both editor and reviewer for comments. First based on your constructive comment we have improved the justification for why we focused on lactating mothers accordingly. The introduction section Line 23-30 Page 1. Second, even though we considered some publications on anemia among reproductive age women ,our primary intention was focusing on the studies conducted on anemia among lactating mothers. However, your comment strongly convinced as. Therefore, we added the literatures on women of reproductive age like “ Kibret KT, Chojenta C, D’Arcy E, Loxton D. Spatial distribution and determinant factors of anaemia among women of reproductive age in Ethiopia: a multilevel and spatial analysis. BMJ open. 2019;9(4):e027276 and Ejigu BA, Wencheko E, Berhane K. Spatial pattern and determinants of anaemia in Ethiopia. PloS one. 2018;13(5” and others.

Comment 3: Data source, sampling design and setting needs to be rewrite in appropriate manner. Whereas measurements should be different sections and can be written accordingly. It would have been better if researchers would have provided region wise sample over the three study period. 

Response 3: Thank you reviewer and editor for your constructive comment. We really appreciated the comment and totally converted sample to region wise approach since the further analysis was region based it is good to provide region based sampling information for the data set. Method section Line 100, page 4. In addition the “Data measurement” was provided in separate section as per your comment and data source, sampling design and setting were strengthened accordingly.

Comment 4: “As the author mentioned “The haemoglobin level was measured for those eligible mothers after having consent and it was adjusted for altitude”. This needs to be clarified as DHS data for the country provide measured haemoglobin level and the adjusted haemoglobin for altitude. So, which one indicator authors have used in the study. The cut off used in this study has not been mentioned in the methodology section.”

Response 4: Thanks reviewer and editor. This section was rewritten again and corrected as follows. “In Ethiopia, Data collection on anemia as key indicator in national surveys was started in 2005 survey and continued in the next two surveys (2011 and 2016). In all surveys , the hemoglobin level was measured for those eligible mothers after having consent and it was adjusted for altitude (18-20). Therefore, the current study was based on the altitude adjusted haemoglobin level which was already provided in the EDHS data. Finally, the women with hemoglobin level <120 g/L were considered as anemic. “Method section Line 89-94, page 4. This cut off point was used from WHO definition of anemia for non-pregnant women as it was indicated in background section. 

Comment 5: Spatial analysis section starts with autocorrelation analysis to spatial scan statistical analysis. Here, the concern is that in many section, authors have provided only the theoretical notions of such the method but missed to elaborate in accordance and need of the undertaken study. Like, how SAC has been executed in the study has found missing. Second, section on SAC mentioned somewhere “leads to rejection of null hypothesis and indicates the presence of spatial correlation”. Does the author have posed research question in same way?

Response 5: Thanks reviewer and editor. We have rewritten this section under heading “Spatial analysis” just by avoiding the subsections. We paraphrased each section as a single paragraph in coherent way which shows the sequence of the spatial analysis methods conducted. We again indicated the purpose of conducting each analysis methods in relation with the theoretical notion of the methods. The second point here was the concept of the following sentence. “Leads to rejection of null hypothesis and indicates the presence of spatial correlation”. Does the author have posed research question in same way?” This sentence was used just to indicate that how the spatial autocorrelation works. When we conduct spatial autocorrelation analysis generally the null hypothesis was random distribution of the disease of interest over the study area. Therefore when we get significant global Moran’s index in spatial autocorrelation analysis result we reject the null hypothesis and deal with spatial clustering to detect further the local level clusters by using other analytic methods like hotspot analysis and spatial scan statistics. It was incorporated to indicate this general concept. However we have rephrased in the revised manuscript to avoid such confusion. Method section Line 106-145 Page 6

Comment 6: Hot spot analysis (HSA) using Gettis-OrdGi* statistics is a appropriate way to check the variation in spatiality across the study areas but missed to the process undertaken in the study. It would be better if authors had been reviewed available study for spatial analysis of anaemia in Ethiopia by Kelemu T.K, et (2019) study using latest wave of Ethiopia DHS data. So, author is advised to go through the paper and restructure the section appropriately.

Response 6: Thanks reviewer for your commitment just to improve our manuscript by searching such relevant studies. I hope we have benefited a lot to strengthen the manuscript. Therefore we have restructured the hotspot analysis and even other sections accordingly as I have described in response 5.

Comment 7: Overall methodology section needs to tightened and strengthened 

Response 7: Thanks reviewer. Based on recommendation we go through the method section organized it as it was indicated in the track change document.

Comment 8: Results are very poorly interpreted, seems researchers are in hurry. So, results sections had to be written in well manner. Figures 3 can be interpreted region wise and changes over period would be better idea. Interpretation in figure heading is not the academic practices, so author is suggested to follow the rigour and pattern/outline to prepare the manuscript. 

Response 8: Thank you reviewer for your valuable comment. Based on your comment the result section is rewritten again. We have avoided the interpretations in the figure heading and adjusted it according to the guideline in all figures. The source for the shape file for maps was also indicated at the figure heading.

Comment 9: It would have been better to elaborate the section on emerging hotspot areas based on the data available for the areas before doing interpolation analysis to detect the primary and secondary clusters.

Response 9: thanks reviewer. We provided the prominent risk areas based on the spatial scan statistics results and the spatial interpolation was conducted to detect the predicted risk areas for anemia in unsampled areas.

Comment 10: In spatial scan analysis, authors have tires to provide glimpse of significant clusters separately for each survey period but it they can provide reason wise prominent clusters, it would be better for planning and programme intervention at the regional level within Ethiopia. 

Response 10: Thanks reviewer since the study involved the three surveys separately we were ended up with a number significant clusters. As per your comment currently we have focused on the most likely clusters only in each survey. As you justified in your comment, even the purpose of conducting spatial scan statistics than the hotspot analysis is to detect most prominent cluster for intervention especially in resource limited areas. Therefore, we welcomed your really constructive comment and acted accordingly.

Comment 11: Table 3 does not have catchy and reliable approach. It would have been better if the researchers would have been mentioned detailed with name or within each of the cluster with maximum and minimum concentration of anaemic lactating mothers. Second, researcher can match for enumeration areas which were found common during the period of 2005, 2011 and 2016, cluster wise. The format of table 3 provided in the text can be produced as supplementary table or appendix table

Response 11: thanks reviewer. Here again we appreciated your comment and reorganized the table by focusing on most likely clusters across each survey. However, we fail to provide the enumeration areas which were found common during three surveys. This was because the enumeration areas used in three periods were different in their number and type as it was indicated in the method section.

Comment 12: Discussion section needs to rewrite as it is just findings of study only in very limited way. It is important to discuss the findings of the study with other previous and existing studies on the issues. 

Response 12: thanks reviewer as per your recommendation we reorganized the discussion in line with the adjustments made in result section. 

Comment 13: In the conclusion section authors have mentioned “Findings suggest that giving priority attentions would be important on water, and other nutrition-related interventions on the identified hotspot areas to prevent and control anaemia incidence among lactating mothers”. The concern is that from where these findings emerged as it is not found anywhere and anyway in the manuscript. 

Response 13: thank you reviewer for your wonderful comment. We have completely avoided this sentence and organized accordingly. Line 289-298 page 12.

Comment 14: Maps and figures are also not readable as are very hazy. So authors are also advised to rework for figure and maps to make it clear before submitting it the journal. 

Response 14: thank you reviewer. We reworked all the figures and maps carefully. We hope all they are readable and clear in the revised manuscript.

Over all the comments were wonderful and we learn a lot from the comments. 

Version two

 Issues raised by academic editor

Thank you editor once again for critical consideration of our manuscript. Here is to declare that we have adjusted the following points based on your recommendation. We also reflected accordingly on the issues which require further clarification.

1. We have checked that our manuscript meets the plos one requirement

2. We have rephrased to avoid the text overlap with the previous publications.

3. We have included the potential limitations of the study in the discussion section.

4. Thank you editor for the constructive comments. Here we are interested to clarify the source of the figures. We kindly declare that all the figures and maps are our own works. The shape file for those maps was accessed from Ethiopian Central Statistical Agency and the geographic coordinates were accessed from measure DHS after being authorized and registered user. Therefore, we prepared all the maps used to show the spatial analysis in each survey by ArcGIS10.7 software. We indicated the source of the shape file at the right lower corner of each map and for more clarity we have also included the source of the shape file at the figure heading in all figures.

5. I the crospondng author included the ORCID in the revised submission.

---

## [Decision Letter · Decision Letter 1]

7 May 2020

PONE-D-19-30322R1

Spatial patterns of anemia among lactating mothers in Ethiopia: Data from Ethiopian Demographic and Health Surveys (2005, 2011 and 2016)

PLOS ONE

Dear Mr liyew,

Thank you for submitting your manuscript to PLOS ONE. After careful consideration, we feel that it has merit but does not fully meet PLOS ONE’s publication criteria as it currently stands. Therefore, we invite you to submit a revised version of the manuscript that addresses the points raised during the review process.

We would appreciate receiving your revised manuscript by Jun 21 2020 11:59PM. To enhance the reproducibility of your results, we recommend that if applicable you deposit your laboratory protocols in protocols.io, where a protocol can be assigned its own identifier (DOI) such that it can be cited independently in the future. For instructions see: http://journals.plos.org/plosone/s/submission-guidelines#loc-laboratory-protocols

We look forward to receiving your revised manuscript.

Kind regards,

William Joe

Academic Editor

PLOS ONE

Reviewers' comments:

Reviewer's Responses to Questions

**Comments to the Author**

1. If the authors have adequately addressed your comments raised in a previous round of review and you feel that this manuscript is now acceptable for publication, you may indicate that here to bypass the “Comments to the Author” section, enter your conflict of interest statement in the “Confidential to Editor” section, and submit your "Accept" recommendation.

Reviewer #1: All comments have been addressed

Reviewer #2: All comments have been addressed

2. Is the manuscript technically sound, and do the data support the conclusions?

Reviewer #1: Partly

Reviewer #2: Yes

3. Has the statistical analysis been performed appropriately and rigorously? 

Reviewer #1: No

Reviewer #2: Yes

4. Have the authors made all data underlying the findings in their manuscript fully available?

Reviewer #1: No

Reviewer #2: Yes

5. Is the manuscript presented in an intelligible fashion and written in standard English?

Reviewer #1: No

Reviewer #2: Yes

6. Review Comments to the Author

Reviewer #1: Spatial Patterns of Anemia among Lactating Mothers in Ethiopia: Data from Ethiopian Demographic and Health Surveys (2005, 2011 and 2016)

Although, the revised paper shows improvement than the previous one but, still there is need to work on it to improve the quality and strength of the paper to meet the criteria of the said journal. Important question is why the spatial hotspot is in those areas. Using data for three points of time to make it comparable also needs logically justification. Some other important comments are as follows-

1. Keep the use of term consistent- like somewhere author used maternal, somewhere lactating and somewhere breastfeeding mothers.

2. Line no 50 are not well connected with line no 49.

3. Authors need to rewrite the data source and sampling section and data measurement section. As these two are overlapping.

4. Authors missed to include anaemia measurement method in Ethiopia and the cut off as well used in the study for lactating mothers.

5. Still results sections needs improvement and there is mismatch on table no, figure no etc. like figure no. 1, 2, 3, 4 etc has mentioned twice with two different headings.

6. Figures are missing at all in the revised edited version.

7. Authors are proving only spatial hotspot and cold spot of the anaemic lactating women, if I am not wrong. But, important is why these are so, is important question and can be worth of the paper.

8. Maps and some tables, figures are missing at all in the revised paper.

9. Authors can have look on the paper related to their concerned issues. https://journals.plos.org/plosone/article?id=10.1371/journal.pone.0197171

Thank you!!!

Reviewer #2: The authors have tried to address the issues raised in the previous review. However, authors still need to expand the discussion section by providing more explanation for the results and relating findings to previuos studies.

Also, weaknesses and strengths stated in the discussion section should be moved to a seperate section titled "Strengths and weaknesses"

7. PLOS authors have the option to publish the peer review history of their article (what does this mean?). If published, this will include your full peer review and any attached files.

Reviewer #1: Yes: Rajesh Raushan

Reviewer #2: No

---

## [Author Response · Author response to Decision Letter 1]

20 May 2020

Date May 20/2020

To: PLOS ONE Journal 

Subject Submitting Revised Manuscript after Reviewers comment 

Manuscript title: Spatial Patterns of Anemia among Lactating Mothers in Ethiopia: Data from Ethiopian Demographic and Health Surveys (2005, 2011 and 2016)

Manuscript ID: PONE-D-19-30322R1

Dear editor/reviewers 

We would like to thank you for your constructive comments on our manuscript. We really appreciate your continuous effort and academic commitment to strengthen our paper. Our point-by-point responses for each comment and questions are described as follow. The details of these changes were shown by track changes feature attached. 

Point by point response for reviewers comment

For reviewer # 1

1. Why the spatial hotspot is in those areas. Authors are proving only spatial hotspot and cold spot of the anemic lactating women, if I am not wrong. But, important is why these are so, is important question and can be worth of the paper.

Author’s response: Thank you. We are very pleased to respond to this comment which is really important point that we missed in previous reflection. 

Though iron supplementation is an essential service for prevention of anemia among lactating mothers, its coverage is very low in these pastoral regions (Afar and Somali) which might be responsible for a high prevalence and hotspot areas of anemia in these regions as compared to other parts of Ethiopia. In addition, the compliance of lactating mothers to the use of iron supplementation survice in these regions is below the recommended level (43). This might lead to an increased risk of anemia among lactating mothers in Afar and Somali regions compared to other regions. The other possible reason behind the consistent hotspot areas in these regions could be nutritional problems such as lack of diatery diversity, consumption of camel and cow milk which had relatively low iron content and the seasonal variation of food they consume (pastoral community). We have included this justification in the revised document (see line 393-401 of the discussion section). 

2. Using data for three points of time to make it comparable also needs logically justification. 

Author’s response: To date, the risk areas (hot spot) of anemia among lactating mothers in Ethiopia are not identified. Thus, this study aimed to explore the spatial pattern of anemia among lactating mothers in Ethiopia over the last one and half-decades at the national level to point out whether there was either the shift or improvement in anemia risk areas following intervention programs in between the survey periods in Ethiopia. Therefore, detecting the geographic variation of anemia during lactation is important to prioritize and design targeted intervention programs to reduce anemia especially in those areas with consistently higher risk of anemia over time (see line 80-86 section of introduction part).

3. Keep the use of term consistent- like somewhere author used maternal, somewhere lactating and somewhere breastfeeding mothers.

Author’s response: Thank you reviewer. We have modified accordingly in the revised manuscript (as you can see the track change feature). 

4. Line no 50 are not well connected with line no 49. 

Author’s response: thank you. It is corrected accordingly

5. Authors need to rewrite the data source and sampling section and data measurement section. As these two are overlapping.

Author’s response: Thank you. We have reorganized this section in logical and sequential approach 

6. Authors missed to include anemia measurement method in Ethiopia and the cutoff point as well used in the study for lactating mothers.

Author’s response: Thank you. We have included the measurement of anemia in the revised manuscript (see line 143 -149 of method section).

7. Still results sections needs improvement and there is mismatch on table no, figure no etc. like figure no. 1, 2, 3, 4 etc has mentioned twice with two different headings.

Author’s response: Thanks. Sorry for the inconvenience we made. We have corrected accordingly and included in the revised manuscript. 

8. Maps and some tables, figures are missing and some are not visible at all in the revised paper.

Author’s response: thanks you reviewer. We have reworked all the maps, figures and tables to make them visible. However, as you have observed in the previous two submissions, when we prepare the multi panel figure using GIMP software the quality and visibility of the figure diminishes. Therefore, in this revision we have prepared the figures separately and labeled like Fig 5(a), Fig 5(b), Fig 5(c) e.t.c. 

9. Has the statistical analysis been performed appropriately and rigorously?

Authors response: We reanalyzed the data rigorously to make the results more visible and readable using appropriate spatial analytic tool. For this purpose we presented each figure separately ( see the revised manuscript). 

10. Have the authors made all data underlying the findings in their manuscript fully available?

Authors response: All the data underlying the findings were fully available in the manuscript. We declare that the authors did not have any special access privileges that others would not have. The data are publicly available upon reasonable request of the DHS MEASURE website through archive@measuredhs.com. after being authorized user. We also included this text in the revised manuscript. 

11. Is the manuscript presented in an intelligible fashion and written in Standard English? 

Authors response: we reedited the whole manuscript by consulting senior English language professionals in our university to improve the grammatical quality of the article and to meet the journal standard. 

For reviewer # 2

1. Authors still need to expand the discussion section by providing more explanation for the results and relating findings to previous studies.

Authors’ response: thank you. We have incorporated your suggestion in the revised manuscript. see line 393-401 of the discussion section and line number _366 -370 of this section. 

2. Also, weaknesses and strengths stated in the discussion section should be moved to a seperate section titled "Strengths and weaknesses"

Authors’ response: We put it in separate section in the revised manuscript. 

Thank you in advance for your constructive comments!!!

---

## [Decision Letter · Decision Letter 2]

16 Jun 2020

PONE-D-19-30322R2

Spatial Patterns of Anemia among Lactating Mothers in Ethiopia: Data from Ethiopian demographic And Health Surveys (2005, 2011,and 2016)

PLOS ONE

Dear Dr. liyew,

Thank you for submitting your manuscript to PLOS ONE. After careful consideration, we feel that it has merit but does not fully meet PLOS ONE’s publication criteria as it currently stands. Therefore, we invite you to submit a revised version of the manuscript that addresses the points raised during the review process.

We look forward to receiving your revised manuscript.

Kind regards,

William Joe

Academic Editor

PLOS ONE

Reviewers' comments:

Reviewer's Responses to Questions

**Comments to the Author**

1. If the authors have adequately addressed your comments raised in a previous round of review and you feel that this manuscript is now acceptable for publication, you may indicate that here to bypass the “Comments to the Author” section, enter your conflict of interest statement in the “Confidential to Editor” section, and submit your "Accept" recommendation.

Reviewer #1: All comments have been addressed

Reviewer #2: All comments have been addressed

2. Is the manuscript technically sound, and do the data support the conclusions?

Reviewer #1: Partly

Reviewer #2: Yes

3. Has the statistical analysis been performed appropriately and rigorously? 

Reviewer #1: Yes

Reviewer #2: Yes

4. Have the authors made all data underlying the findings in their manuscript fully available?

Reviewer #1: Yes

Reviewer #2: Yes

5. Is the manuscript presented in an intelligible fashion and written in standard English?

Reviewer #1: No

Reviewer #2: Yes

6. Review Comments to the Author

Reviewer #1: Reviewer Report

Spatial Patterns of Anemia among Lactating Mothers in Ethiopia: Data from Ethiopian Demographic and Health Surveys (2005, 2011 and 2016)

1. In the conclusion of abstract section- Therefore, public health intervention activities designed in a targeted approach to impact high-risk populations as well as the geographic regions is vital to narrow anemia disparity in Ethiopia’- needs attention for correction the statement ‘is vital to narrow anemia disparity in Ethiopia’.

2. To date, the risk areas (hot spot) of anemia among lactating mothers in Ethiopia are not identified- so what are the other studies related to anaemia among lactating mothers would be important to justify the statement (Line 91).

3. Line No 115-120 is not required in the way it has been written. Authors are advised to look for section on data source from various published articles in journal of high repute.

4. In Line number 125, it would be better to mention the age range for ‘reproductive age women’.

5. ‘Data measurement’ heading is not appropriate.

6. In line no 164 as the author has used word ‘location data’. I think those were geo-referencing data. So its always good to use appropriate word.

7. Within the methodology section, spatial analysis section is improved and well written than the previous one.

8. Line no 214-218: “The prevalence of anemia during lactation was intermittently increasing across regions in Ethiopia. Even though it had decreased from 2005 to 2011 almost in all regions, its prevalence increased from 2011 to 2016 in all regions including the two administrative cities. The highest prevalence was observed in the Somali regional state (68%) and the Afar region (47%) in 2016”- The statement should be written very carefully. As line no 214 reflects that since 2005, anaemia among lactating mother is on increase. But, that is not the case as the authors are stating in the line no-215/216 that it had decreased between 2005 and 2011.

9. Don’t use like ‘in the following figure’ or ‘in the above figure/map, tables’ etc, as at many place in the manuscript, it has written like that,

10. As in table 2, keep number of digit identical after the decimal. As somewhere its two digit, somewhere, its three digit in second column. In column 4, keep three digits after decimal. Keep it identical for all the tables.

11. Line no 253- Can be written as all three consecutive survey periods. Because, all surveys has different meaning.

12. Line no 264- 270 needs to be written in well manner as the- In the above figure, HH (High-High) means high rates of anemia surrounded by similar characteristics; HL (High-Low) means high rates of anemia surrounded by low rates of anemia; LH (Low-High) means low rates of anemia cases surrounded by high rates of anemia cases; and LL (Low-Low) means low rates of anemia cases surrounded by similar characteristics. The red (HH) color indicates hotspot areas of anemia, the dark blue (LL) color indicates cold spot areas of anemia, and the dark yellow (HL ) and yellow (LH) colors indicate the outliers.-. Section needs rewrite as writing HL means…., LL means is not the standard writing style in result sections. This is not the right way to interpret the map. The authors can narrate what is emerging out from the said map, not like HL means High-Low. As the Map 5a, 5b, 5c are for three different time period so it would be better to do compare the regions. The colour combination should be part of methodology section.

13. Authors mentioning hot spot, cold spot etc in figure 4,5 6. Which needs proper interpretation like what are those spots, is there any change during the three survey period as authors have written about in very brief in discussion section.

14. ‘This implies, that the special attention of policy makers for anemia reduction should be in those high-risk areas of the country’- this statement needs correction/modification as per the study objectives.

15. ‘This spatial heterogeneity of anemia clustering was again observed prominently in Afar and Somali regions. This showed that the spatial clustering of anemia is more or less consistently higher in the Afar region in all EDHS surveys and the Somali region in the latest two surveys (2011 and 2016)’- Validate your findings with other available studies or is this new finding emerged from your study for the first time.

16. Policy suggestion or public health measures are missing, authors can think on those lines.

Overall, there is improvement in the paper than the previous one, but the scientific rigor in interpreting the results emerging from the map is somewhat missing. Discussion section still needs to strengthen. It would be better to reduce the number of maps wherever it’s possible. The paper still needs English editing as in the data and methodology section flow and consistency was missing as well. Authors can also think about the title of the paper.

Thank You!!!

Reviewer #2: (No Response)

7. PLOS authors have the option to publish the peer review history of their article (what does this mean?). If published, this will include your full peer review and any attached files.

Reviewer #1: Yes: Rajesh Raushan

Reviewer #2: No

---

## [Author Response · Author response to Decision Letter 2]

24 Jun 2020

Date Jun 24/2020

To: PLOS ONE Journal 

Subject Submitting Revised Manuscript after Reviewers comment 

Manuscript title: Spatiotemporal Patterns of Anemia among Lactating Mothers in Ethiopia using data from Ethiopian Demographic and Health Surveys (2005, 2011 and 2016)

Manuscript ID: PONE-D-19-30322R2

Dear editor/reviewers 

We would like to thank you for your constructive comments on our manuscript. We really appreciate your continuous effort and academic commitment to strengthen our paper. Our point-by-point responses for each comment and questions are described as follow. The details of these changes were shown by track changes feature attached. 

Point by point response for reviewers comment

For reviewer # 1

1. In the conclusion of abstract section- Therefore, public health intervention activities designed in a targeted approach to impact high-risk populations as well as the geographic regions is vital to narrow anemia disparity in Ethiopia’- needs attention for correction the statement ‘is vital to narrow anemia disparity in Ethiopia’. 

Author’s response: thank you reviewer. We corrected accordingly. (See the abstract section line 41-42)

2. To date, the risk areas (hot spot) of anemia among lactating mothers in Ethiopia are not identified- so what are the other studies related to anemia among lactating mothers would be important to justify the statement (Line 91).

Author’s response: thanks reviewer we have provided additional citations regarding studies conducted on determinants of anemia among lactating mothers. (We kindly request to see introduction section; line 81-82)

3. Line No 115-120 is not required in the way it has been written. Authors are advised to look for section on data source from various published articles in journal of high repute.

Author’s response: Thank you reviewer. We have modified accordingly in the revised manuscript (as you can see the track change feature). 

4. In Line number 125, it would be better to mention the age range for ‘reproductive age women’

Author’s response: thank you reviewer we specified it as “reproductive age women (15-49)” 

5. Data measurement’ heading is not appropriate.

Author’s response: Thank you. Since this section describes about how anemia is measured which is our problem of interest. Therefore we changed “data management” to “outcome variable” (see line 129)

6. In line no 164 as the author has used word ‘location data’. I think those were geo-referencing data. So it’s always good to use appropriate word

Author’s response: Thank you. We have corrected accordingly. (see line 124 of method section).

7. Within the methodology section, spatial analysis section is improved and well written than the previous one

 Author’s response: Thanks reviewer for your critical review.

8. Line no 214-218: “The prevalence of anemia during lactation was intermittently increasing across regions in Ethiopia. Even though it had decreased from 2005 to 2011 almost in all regions, its prevalence increased from 2011 to 2016 in all regions including the two administrative cities. The highest prevalence was observed in the Somali regional state (68%) and the Afar region (47%) in 2016”- The statement should be written very carefully. As line no 214 reflects that since 2005, anemia among lactating mother is on increase. But, that is not the case as the authors are stating in the line no-215/216 that it had decreased between 2005 and 2011.

Author’s response: thanks you reviewer. We have reinterpreted this section.(we kindly request to see the track change feature)

 9. Don’t use like ‘in the following figure’ or ‘in the above figure/map, tables’ etc, as at many place in the manuscript, it has written like that

Author’s response: we totally removed such phrases per your comment (see the revised manuscript). 

10. As in table 2, keep number of digit identical after the decimal. As somewhere its two digit, somewhere, its three digit in second column. In column 4, keep three digits after decimal. Keep it identical for all the tables

Authors response: thank you reviewer we corrected in the revised manuscript. (See the revised manuscript). 

11. Line no 253- Can be written as all three consecutive survey periods. Because, all surveys has different meaning

Authors’ response: thanks reviewer it is corrected accordingly.

12. Line no 264- 270 needs to be written in well manner as the- In the above figure, HH (High-High) means high rates of anemia surrounded by similar characteristics; HL (High-Low) means high rates of anemia surrounded by low rates of anemia; LH (Low-High) means low rates of anemia cases surrounded by high rates of anemia cases; and LL (Low-Low) means low rates of anemia cases surrounded by similar characteristics. The red (HH) color indicates hotspot areas of anemia, the dark blue (LL) color indicates cold spot areas of anemia, and the dark yellow (HL ) and yellow (LH) colors indicate the outliers.-. Section needs rewrite as writing HL means…., LL means is not the standard writing style in result sections. This is not the right way to interpret the map. The authors can narrate what is emerging out from the said map, not like HL means High-Low. As the Map 5a, 5b, 5c are for three different time period so it would be better to do compare the regions. The color combination should be part of methodology section.

Authors’ response: thank you. We have reinterpreted the results from cluster and outlier analysis (we kindly request to see the track change feture)

13. Authors mentioning hot spot, cold spot etc in figure 4,5 6. Which needs proper interpretation like what are those spots, is there any change during the three survey period as authors have written about in very brief in discussion section

Authors’ response: thanks reviewer we improved the interpretation.( see line 211-217 of result section)

14. This implies, that the special attention of policy makers for anemia reduction should be in those high-risk areas of the country’- this statement needs correction/modification as per the study objectives.

Authors’ response; thanks reviewer we critically modified this statement. (See line 266-274 section of discussion)

15. This spatial heterogeneity of anemia clustering was again observed prominently in Afar and Somali regions. This showed that the spatial clustering of anemia is more or less consistently higher in the Afar region in all EDHS surveys and the Somali region in the latest two surveys (2011 and 2016)’- Validate your findings with other available studies or is this new finding emerged from your study for the first time.

Authors’ response; thank you reviewer. As to our knowledge concerned it is new finding. Thus, it is a little bit difficult to validate with other studies. 

16.Policy suggestion or public health measures are missing, authors can think on those lines.

Authors’ response; thank you reviewer. We really appreciate the comment. We included the policy implication of the findings in the revised document.(we kindly request to see the line 294-299 section of discussion)

17. Overall, there is improvement in the paper than the previous one, but the scientific rigor in interpreting the results emerging from the map is somewhat missing. Discussion section still needs to strengthen. It would be better to reduce the number of maps wherever it’s possible. The paper still needs English editing as in the data and methodology section flow and consistency was missing as well. Authors can also think about the title of the paper.

Authors’ response; thank you reviewer. We improved the interpretation, Discussion and English editing issues over all readability of the article in the revised manuscript as it can be observed from the track changes.

Thank you in advance for your constructive comments!!!

---

## [Decision Letter · Decision Letter 3]

22 Jul 2020

Spatiotemporal Patterns of Anemia among Lactating Mothers in Ethiopia using data from Ethiopian demographic And Health Surveys (2005, 2011,and 2016)

PONE-D-19-30322R3

Dear Dr. liyew,

We’re pleased to inform you that your manuscript has been judged scientifically suitable for publication and will be formally accepted for publication once it meets all outstanding technical requirements.

Kind regards,

William Joe

Academic Editor

PLOS ONE

Additional Editor Comments (optional):

Reviewers' comments:

Reviewer's Responses to Questions

**Comments to the Author**

1. If the authors have adequately addressed your comments raised in a previous round of review and you feel that this manuscript is now acceptable for publication, you may indicate that here to bypass the “Comments to the Author” section, enter your conflict of interest statement in the “Confidential to Editor” section, and submit your "Accept" recommendation.

Reviewer #1: All comments have been addressed

Reviewer #2: All comments have been addressed

2. Is the manuscript technically sound, and do the data support the conclusions?

Reviewer #1: Yes

Reviewer #2: Yes

3. Has the statistical analysis been performed appropriately and rigorously? 

Reviewer #1: Yes

Reviewer #2: Yes

4. Have the authors made all data underlying the findings in their manuscript fully available?

Reviewer #1: Yes

Reviewer #2: Yes

5. Is the manuscript presented in an intelligible fashion and written in standard English?

Reviewer #1: Yes

Reviewer #2: Yes

6. Review Comments to the Author

Reviewer #1: Minor suggestions

1. Line no 127- Is four survey conducted or typed by mistake. I think it would be three surveys.

2. Line no- 208 can be written as Even though prevalence of Anaemia had decreased between 2005 and 2011.

3. Authors have used somewhere one and half decade. I think from 2005 to 2016, its 11 years. So, please do correct it.

Overall, the revised version seems improved and looks impressive and readable. So authors can take care of minor typo-error and other necessities of the PLOS ONE.

Thank You!!!

Reviewer #2: (No Response)

7. PLOS authors have the option to publish the peer review history of their article (what does this mean?). If published, this will include your full peer review and any attached files.

Reviewer #1: **Yes: **Rajesh Raushan, PhD

Reviewer #2: No

---

## [Editor Report · Acceptance letter]

24 Jul 2020

PONE-D-19-30322R3 

Spatiotemporal Patterns of Anemia among Lactating Mothers in Ethiopia using data from Ethiopian Demographic and Health Surveys (2005, 2011 and 2016) 

Dear Dr. Liyew:

I'm pleased to inform you that your manuscript has been deemed suitable for publication in PLOS ONE. Congratulations! Your manuscript is now with our production department. 

Kind regards, 

on behalf of

Dr. William Joe 

Academic Editor

PLOS ONE